# γ−MoD: Exploring Mixture-of-Depth Adaptation for Multimodal Large Language Models

Yaxin Luo[1], Gen Luo[2†], Jiayi Ji[3,4], Yiyi Zhou[3], Xiaoshuai Sun[3], Zhiqiang Shen[1], Rongrong Ji[3]

[1]MBZUAI    [2]OpenGVLab, Shanghai AI Laboratory
[3]Xiamen University    [4]National University of Singapore

**Project Page:** Gamma-MOD

## ABSTRACT

Despite the significant progress in multimodal large language models (MLLMs), their high computational cost remains a barrier to real-world deployment. Inspired by the mixture of depths (MoDs) in natural language processing, we aim to address this limitation from the perspective of "activated tokens". Our key insight is that if most tokens are redundant for the layer computation, then can be skipped directly via the MoD layer. However, directly converting the dense layers of MLLMs to MoD layers leads to substantial performance degradation. To address this issue, we propose an innovative MoD adaptation strategy for existing MLLMs called γ-MoD. In γ-MoD, a novel metric is proposed to guide the deployment of MoDs in the MLLM, namely *rank of attention maps* (ARank). Through ARank, we can effectively identify which layer is redundant and should be replaced with the MoD layer. Based on ARank, we further propose two novel designs to maximize the computational sparsity of MLLM while maintaining its performance, namely *shared vision-language router* and *masked routing learning*. With these designs, more than 90% dense layers of the MLLM can be effectively converted to the MoD ones. To validate our method, we apply it to three popular MLLMs, and conduct extensive experiments on 9 benchmark datasets. Experimental results not only validate the significant efficiency benefit of γ-MoD to existing MLLMs but also confirm its generalization ability on various MLLMs. For example, with a minor performance drop, *i.e.,* -0.9%, γ-MoD can reduce the training and inference time of LLaVA-HR by 31.0% and 53.2%, respectively.

## 1 INTRODUCTION

Recent years have witnessed the great success of large language models (LLMs) in natural language processing (NLP) (Achiam et al., 2023; Touvron et al., 2023; Cai et al., 2024b), which attracts increasing attentions in extending LLMs to vision-language (VL) tasks. Despite the progress, recent multimodal large language models (MLLMs) (Liu et al., 2024d;c; Chen et al., 2024a; Alayrac et al., 2022) are often criticized by their expensive computational costs. For example, the inference speed of existing MLLMs like LLaVA-HR (Luo et al., 2024) is still far from practical requirements, *e.g.,* 4.7 samples per second. Driven by the progress of NLP, recent advances have employed the mixture-of-experts (MoEs) (Lin et al., 2024a; Jiang et al., 2024) to MLLMs to reduce the "activated parameters", thus achieving trade-off between efficiency and performance.

Orthogonal to MoEs, we aim to tackle the efficiency bottleneck of MLLMs from the perspective of "activated tokens". As shown in Fig. 1 (a), a large number of tokens are less important in the computation, such as visual background and prepositional words. However, existing MoEs still allocate the same experts to all input tokens, leading to redundant computational costs. A promising solution to this issue is the recently proposed mixture-of-depths (MoDs) in NLP (Raposo et al., 2024), which equips each token with a router to determine whether a module should be computed.

---

†Corresponding author.

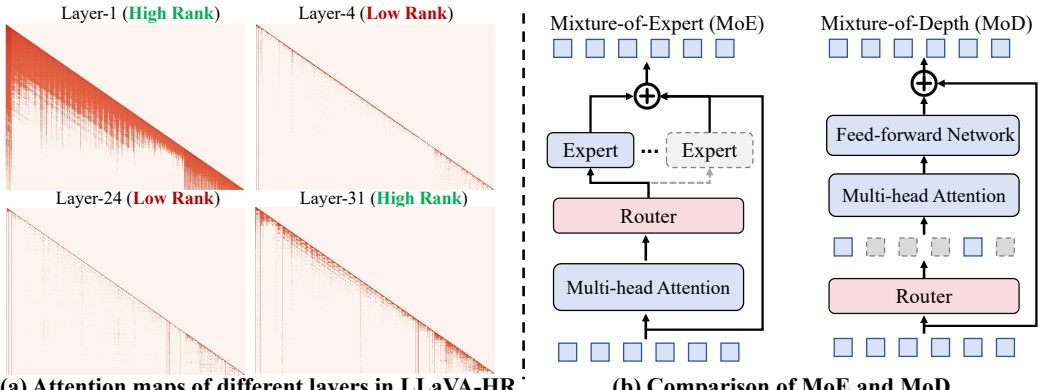

Figure 1: **Visualization of attention maps in the MLLM and comparison of MoE with MoD.** (a) Lower-rank layers often exhibit redundancy in their attention computation. (b) Different from MoE, MoD achieves the computational sparsity from the perspective of "activated token", where the computational budget is dynamically allocated to each token.

However, recent MoDs (Raposo et al., 2024) typically require pre-training LLMs from scratch, and their employment on MLLMs still remains under-explored.

In this paper, we focus on the efficient adaptation of MoDs to existing MLLMs. In particular, our goal is to maximize the computational sparsity of MLLMs while maintaining competitive performance. However, directly converting all dense layers of MLLMs to MoD layers leads to significant performance degradation, *e.g.,* -33.3% of LLaVA-HR (Luo et al., 2024) on TextVQA (Singh et al., 2019). In practice, we observe that such issue is mainly caused by two aspects. Firstly, the deployment of MoDs lacks a practical guidance to measure the layer redundancy, thus undermining the necessary dense layers. As illustrated in Fig. 1 (a), attention patterns vary significantly across layers, and some layers exhibit less redundancy. Additionally, the setting of MLLMs, *e.g.,* input modality, differs substantially from that of LLMs, making the direct adaptation of MoDs suboptimal.

To overcome these limitations, we first propose a novel metric to guide the deployment of MoDs in MLLMs, called the *rank of attention maps* (ARank). Our key insight is that lower-rank attention maps indicate that fewer tokens are necessary for computation. As shown in Fig. 1 (a), most of tokens of *Layer-4* are assigned small attention weights, contributing minimally to the final output. This provides a valuable hint for us to replace the redundant layer with the MoD one under the guidance of ARank. In practice, the calculation of ARank is both efficient and flexible. Empirically, we find that the average ARank always keeps the similar despite the change of samples. Therefore, randomly sampling a small amount of data can already accurately estimate the ARanks.

Based on the ARank, we propose an innovative MoD adaptation strategy for existing MLLMs, called $\gamma$-MoD. Specifically, $\gamma$-MoD is a plug-and-play adaptation approach that can be seamlessly integrated into current MLLMs via instruction tuning. In $\gamma$-MoD, two novel designs are adopted to maximize its benefits to MLLMs, namely *shared vision-language router* and *masked routing learning*. The shared vision-language router performs routing on the entire multimodal sequence and uses a weight-sharing strategy to facilitate optimization. Then, masked routing learning is introduced to prevent critical tokens from being skipped during training, *i.e.,* instruction tokens. With these designs, over 90% of dense layers can be converted to MoD layers with minimal performance sacrifice, resulting in even larger computational sparsity than the native MoD-based LLM (Raposo et al., 2024).

To validate $\gamma$-MoD, we apply it to two popular MLLMs and conduct extensive experiments on 9 vision-language benchmarks. Experimental results show that $\gamma$-MoD significantly improves the training and inference efficiency of existing MLLMs while keeping their performance competitive. For example, $\gamma$-MoD reduces 51.6% Flops, 31% training time and 53.2% inference time for LLaVA-HR (Luo et al., 2024), but its average performance decline is only -1.5%. More importantly, the great generalization ability of $\gamma$-MoD is also witnessed on different MLLM structures and parameter sizes. Overall, the contribution of the paper can be summarized in three folds:

- We present a novel mixture-of-depth (MoD) framework for the sparse computation of existing MLLMs, namely $\gamma$-MoD, which can seamlessly convert most dense layers in MLLMs to the sparse MoD layers.

- We propose an innovative metric to measure the layer redundancy, namely rank of attention maps (ARank). With ARank, we can best determine that which dense layer should be convert to the MoD one.

- We carefully explore the design of $\gamma$-MoD in existing MLLMs, including the shared vision-language router and the masked routing learning, which can achieve up to 51.6% computational sparsity with minor performance sacrifice. Extensive experiments also confirm the generalization ability of $\gamma$-MoD.

## 2 RELATED WORK

### 2.1 MULTIMODAL LARGE LANGUAGE MODELS

Large language models (LLMs) (Achiam et al., 2023; Touvron et al., 2023; Jiang et al., 2024; Almazrouei et al., 2023; Cai et al., 2024b; Abdin et al., 2024; Shen et al., 2023) have proven their strong capabilities in various natural language processing tasks (Paperno et al., 2016; Fyodorov et al., 2000; Reddy et al., 2019; Ziegler et al., 2019). Motivated by this, numerous efforts (Liu et al., 2024d; Bai et al., 2023a; Ye et al., 2023; Dai et al., 2023; Chen et al., 2024b; Li et al., 2024b; Tong et al., 2024; Rasheed et al., 2024; Dong et al., 2023; Xie et al., 2024; Zhou et al., 2024; Chen et al., 2023; Alayrac et al., 2022; Sun et al., 2024) have been devoted into extending LLMs to multimodal large language models (MLLMs). Among them, the most representative work is LLaVA (Liu et al., 2024d), which uses a lightweight project to connect a visual encoder and an LLM. This simple framework has now become the de-facto paradigm in the community, empowering a set of MLLMs like Mini-Gemini (Li et al., 2024b) and InternVL (Chen et al., 2024b). Recently, researchers have shifted their attentions to high-resolution MLLMs. For example, LLaVA-NexT (Liu et al., 2024c) and InternVL-1.5 (Chen et al., 2024a) adopt the dynamic image slicing strategy for high-resolution adaptation. LLaVA-HR (Luo et al., 2024) further propose a dual-branch structure to reduce the cost of high-resolution MLLMs. Despite the effectiveness, existing high-resolution MLLMs (Liu et al., 2024c; Li et al., 2024a) will produce a much longer input tokens, resulting in prohibitively expensive computational costs. In this paper, the proposed $\gamma$-MoD can greatly overcome the efficiency bottleneck of existing MLLMs, which is significant for their practical applications.

### 2.2 SPARSE COMPUTATION FOR LLMS

Recently, an influx of attentions have been focused on the sparse computation of LLMs. Specifically, the mixture of experts (MoEs) are the most popular technology in the community (McKinzie et al., 2024; Cai et al., 2024a; Xue et al., 2024), which dynamically activates part of expert networks for each token, thereby achieving trade-offs between capability and efficiency. For instance, MoE-LLaVA (Lin et al., 2024a) proposed a novel approach to convert a dense MLLM to a mixture-of-expert structure. However, these methods often require additional training costs to realize the adaptation to MLLMs. Orthogonal to MoE, Raposo et al. (2024) proposed the mixture of depths (MoDs) to dynamically allocate computations for each token. Compared to MoE, the main principle of MoD is to reduce the "activated tokens" instead of the "activated parameters". This paradigm has shown great potentials for the sparse computation of LLMs, but its potential on MLLM is still under exploration. Recently, token-based pruning methods have emerged as a new promising solution. The most representative one is the FastV (Chen et al., 2025), which directly deletes the unimportant visual tokens according to their attention scores, thus achieving significant computational savings without compromising performance. Orthogonal to these works, we are the first to explore MoDs on MLLMs, which can seamlessly realize sparse computations of exiting MLLMs on both visual and textual tokens.

## 3 PRELIMINARIES

We first recap the mechanism of *Mixture of Experts* (MoEs) and *Mixture of Depths* (MoDs).

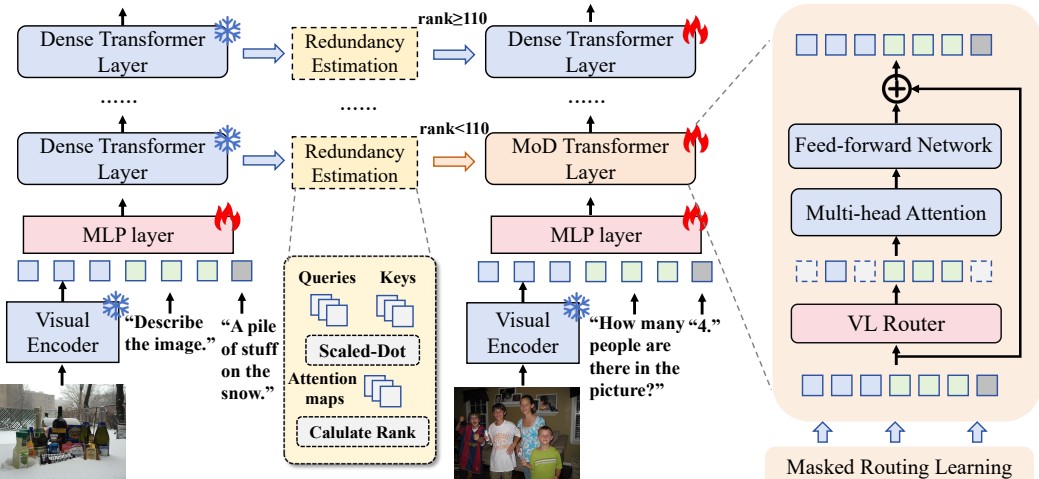

Figure 2: **Illustration of our $\gamma$-MoD adaptation on LLaVA-HR.** $\gamma$-MoD is a plug-and-play approach that can be directly applied in existing MLLMs. After vision-language alignment, $\gamma$-MoD can replace most redundant layers with MoD ones via the rank-based redundancy estimation.

**Mixture of experts.** In particular, the main principle of MoE is to reduce the "*activated parameters*" in dense models. Existing MoE-based LLMs (Dai et al., 2024; Liu et al., 2024a; Lin et al., 2024a; Jiang et al., 2024) and MLLMs (Luo et al., 2024; Chen et al., 2024a; Liu et al., 2024d) often contain multiple FFN modules in their layers, also termed *experts*. During training and inference, only few experts are activated to participate in computations, thus retaining the trade-offs between performance and efficiency. Given input features $x \in \mathbb{R}^{l \times d}$, MoE mechanism can be defined by

$$x = x + \sum_{j=1}^{k} \mathcal{D}_j(x)R_j(x). \tag{1}$$

Here, $\mathcal{D}(\cdot)$ denotes the expert layer, *i.e.,* FFN. $k$ is the number of activated experts, and $R_j(\cdot)$ is the corresponding routing function. In practice, top-k experts are selected according to their routing scores, where $k$ is much smaller than the total number of experts $K$.

**Mixture of depths.** Different from MoEs, MoDs aim to improve the model efficiency via the reduction of "*activated tokens*". Compared to MoEs, the routing mechanism of MoDs performs on input tokens, and most tokens will directly skip the dense layer in MLLMs. Thus, MoDs can be written as

$$x_j = \begin{cases} x_j + \mathcal{D}(x_j)R(x_j) & \text{if } R(x_j) \geq \delta_s, \\ x_j & \text{if } R(x_j) < \delta_s, \end{cases} \tag{2}$$

where $x_j \in \mathbb{R}^d$ denotes the token vector in $x$, and $\delta_s$ is a routing threshold. As defined in Eq. 2, inactive tokens will directly skip the layer $\mathcal{D}(\cdot)$ to save the computational cost.

**Discussion.** In existing MLLMs (Lin et al., 2024a), MoE is typically used to efficiently scale up the model size, while its computations are not directly reduced. In contrast, MoD can perform as a plug-and-play module to save the cost of a common dense layer, which is more significant to the efficient scenario. Unfortunately, the adaptation of MoD to existing MLLMs is still under-explored, and its practical use in LLMs also requires expensive pretraining.

## 4 METHOD

### 4.1 OVERVIEW

In this paper, we propose a novel method to efficiently deploy MoDs to existing MLLMs, namely $\gamma$-MoD. The core principle of $\gamma$-MoD is to identify redundant MLLM layers via a novel metric called

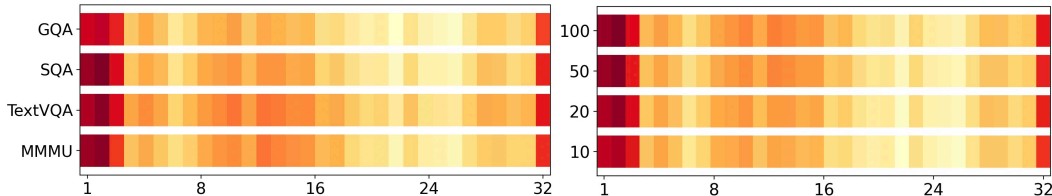

Figure 3: **Visualization of ARank based on different tasks (left) and sample sizes (right).** The horizontal axis represents the layer index of LLaVA-HR. The darker color indicates the larger ARank.

*rank of attention maps* (ARank) and replace them with the proposed MoD layer. Therefore, the deployment of $\gamma$-MoD in the given MLLM, *i.e.,* $\mathcal{F}_{\text{MLLM}}(\cdot)$, can be formulated by

$$\mathcal{F}_{\text{MLLM}} = \mathcal{G}_0 \circ \mathcal{G}_1 \circ \mathcal{G}_2 ... \circ \mathcal{G}_n,$$

$$\text{where} \quad \mathcal{G}_i = \begin{cases} \mathcal{D}_i & \text{if } \tau(\mathcal{D}_i) \geq \delta_\tau, \\ \mathcal{S}_i & \text{if } \tau(\mathcal{D}_i) < \delta_\tau. \end{cases} \tag{3}$$

Here, $\mathcal{G}(\cdot)$ denotes the layer of the MLLM, where $\mathcal{S}(\cdot)$ and $\mathcal{D}(\cdot)$ indicate the dense layer and its MoD alternative, respectively. $\tau(\cdot)$ is a function to estimate the redundancy of the given dense layer $\mathcal{D}_i$, and $\delta_\tau$ is a threshold. Given the architecture in Eq. 3, $\gamma$-MoD aims to maximize the sparsity while maintaining the performance. Thus, the optimization objective of $\gamma$-MoD can be written as:

$$\arg \min_{\theta, \theta_r} \mathcal{L}_{obj}(\mathcal{F}_{\text{MLLM}}(x^0; \theta)) + \sum_{i=1}^{k} \mathcal{L}_{aug}(R(x^i; \theta_r)),$$

$$\text{s.t.} \quad \frac{1}{k \cdot d} \sum_{i=1}^{k} \sum_{j=1}^{d} \mathbb{I}_{R(x_j^i) < \delta_s} = \alpha. \tag{4}$$

Here, $\mathcal{L}_{obj}$ and $\mathcal{L}_{aug}$ denote the auto-regressive loss and the routing loss for the router $R(\cdot)$, respectively. $x^i$ is the input tokens of $i$-th layer, and $\alpha$ is the pre-defined sparse target. $\mathbb{I}_{R(x_j^i) < \delta_s} \rightarrow \{0, 1\}$ is the indicator function, which is equal to 1 when $R(x_j^i) < \delta_s$. And k is the number of layers, d denotes the number of tokens per layer.

## 4.2 RANK-BASED REDUNDANCY ESTIMATION

The key challenge of $\gamma$-MoD is how to identify the dense layer that should be converted to the MoD one. The original MoD-based LLM (Raposo et al., 2024) overcomes this issue by the hand-craft attempt, which is still sub-optimal and time-consuming. However, in existing MLLMs, the LLM is already pre-trained on large scale of corpus, which can intuitively provide sufficient knowledge to achieve the process automatically.

Motivated by this, we propose an innovative metric to estimate the token-wise redundancy of a layer in MLLM, namely *rank of attention maps* (ARank). In particular, given tokens $x^i \in \mathbb{R}^{l \times d}$ of $i$-th layer, ARank is defined by the average rank of attention maps:

$$\tau(x^i, \mathcal{D}_i) = \frac{1}{n_h} \sum_{h=1}^{n_h} \text{rank}(A_h),$$

$$\text{where} \quad A_h = (x^i W_Q^h)(x^i W_K^h)^T. \tag{5}$$

Here, rank$(\cdot)$ denotes the rank calculation. $n_h$ is the number of attention heads. $A_h \in \mathbb{R}^{l \times l}$ is the attention map in $h$-th head, and $W_Q^h \in \mathbb{R}^{d \times \frac{d}{h}}$ and $W_K^h \in \mathbb{R}^{d \times \frac{d}{h}}$ are the corresponding weights.

**Theoretical analysis of ARank.** In Eq. 5, attention map $A_h$ can well reflect the contribution of different tokens. Thus, $A_h$ with a low rank suggests that most tokens are less informative. To validate this, we conduct a SVD (G.H.Goulb & C.Reinsch, 1971) analysis for $A_h$, which is written as

$$A_h = \sum_{i=1}^{r} \sigma_i u_i v_i^T = \sum_{i=1}^{r'} \sigma_i u_i v_i^T + \sum_{i=r'+1}^{r} \sigma_i u_i v_i^T, \tag{6}$$

where $r$ is the rank of $A_h$ and $r' \ll r$ is a constant value. $\sigma_i$, $u_i$ and $v_i$ denote the $i$-th single value, left single vector and right single vector of $A_h$, respectively. As shown in Eq. 6, $A_h$ can be deposed to a matrix of rank $r'$ and additional information, *i.e.,* $\sum_{i=r'+1}^{r} \sigma_i u_i v_i^T$. Therefore, lower-rank attention map suggests higher redundancy, which implies that MoD can be deployed to skip most tokens.

**Practical calculation of ARank.** As defined in Eq. 5, it is still challenging to accurately calculate the ARank due to the variance of individual samples. Inspired by HRank (Lin et al., 2020), we estimate ARank using its expectation on a batch of samples. Different from HRank, we aim to estimate the layer redundancy by the rank of their attention maps, thus guiding the deployment of MoD. Specifically, ARank estimates layer redundancy based on the rank of attention maps, enabling its use in guiding the deployment of MoD. As shown in Fig. 3, we visualize the average ARank values of LLaVA-HR (Luo et al., 2024) across different input samples. These results demonstrate that the expected ARank remains largely consistent across tasks, indicating that a small batch size is sufficient for reliable computation. In our experiments, we set the sample size to 50 to balance computational efficiency and accuracy.

### 4.3 MIXTURE-OF-DEPTH ADAPTATION

To maximize the effectiveness of MoDs to existing MLLMs, we carefully investigate the micro design of MoDs, including the shared vision-language router and the masked routing learning.

**Shared vision-language router.** Conventional MoDs (Raposo et al., 2024) are designed for LLMs, so their routing is only performed on textual tokens. In MLLMs, such a strategy is sub-optimal due to the large redundancy of visual tokens (Jin et al., 2024; Kim et al., 2024). Therefore, the router of $\gamma$-MoD, *i.e.,* $R(\cdot)$, aims to skip both visual and textual tokens, which is defined by

$$R(x) = \text{softmax}(xW_R + b_R), \tag{7}$$

where $x = \{q, a, t\}$ denotes the vision-language tokens, which consist of question tokens $q \in \mathbb{R}^{l_q \times d}$, image tokens $a \in \mathbb{R}^{l_a \times d}$ and textual response tokens $t \in \mathbb{R}^{l_t \times d}$. $W_R \in \mathbb{R}^{l \times 2}$ and $b_R \in \mathbb{R}^2$ are the weights and bias, respectively. Notably, we use a binary softmax function to produce the routing probability, where $R(x)^0$ denotes the probability of skipping. Based on Eq. 7, we further share the router parameters for all MoD layers, which is significant for the stable optimization. To explain, the shared router receives more gradients from different layers, greatly facilitating its convergence at the beginning of training.

**Masked routing learning.** During VL training, not all tokens contribute equally to the optimizing process. In particular, the skip of key tokens in the question, *e.g.,* subject, will greatly hurt the generative training as the answer relies on these conditional elements. Therefore, we introduce a masked routing learning strategy to prevent these tokens from being dropped during training. In this case, the objective of the routing learning can be defined by

$$\mathcal{L}_{aug}(x) = \log\left(R(x)^1 \cdot M_q\right)\hat{R} + \log\left(1 - R(x)^0 \cdot M_q\right)(1 - \hat{R}). \tag{8}$$

Here, $M_q \in \mathbb{R}^{l \times 1}$ denotes the binary mask, where the question tokens are assigned to 0. $\hat{R} \in$ is the one-hot vector, where the position with top-k routing scores are assigned to 1.

**The training scheme.** Typically, MLLM training is divided into two stages: vision-language (VL) alignment and instruction tuning. $\gamma$-MoD is a plug-and-play adaptation method, which is deployed in the instruction tuning stage. Therefore, we can skip the VL alignment by directly using the well pre-trained projector. Then, $\gamma$-MoD then evaluates layer redundancy using the ARank metric and replaces redundant layers with MoD layers. During instruction tuning, the routing parameters are jointly optimized via the routing and task objectives. Importantly, all other training configurations can remain consistent with the original MLLM setup, ensuring seamless integration of $\gamma$-MoD.

## 5 EXPERIMENTS

### 5.1 DATASETS AND METRICS

We evaluate our $\gamma$-MoD on five MLLM benchmarks, which includes POPE (Li et al., 2023), MME (Fu et al., 2024), MMB (Liu et al., 2024e), MMMU (Yue et al., 2024) and MM-Vet (Yu et al., 2023). We

Table 1: **Comparison of different $\gamma$-MoD configurations on LLaVA-HR.** The default setting used in the table is colored in gray. "Q" and "A" refer to question and answer tokens.

| Methods | GQA | | SQA | | MMMU | | TextVQA | | Average | | |
|---|---|---|---|---|---|---|---|---|---|---|---|
| | Acc. | Skip | Acc. | Skip | Acc. | Skip | Acc. | Skip | Acc. | TFlops | Skip |
| LLaVA-HR (Luo et al., 2024) | 64.2 | 0% | 67.9 | 0% | 34.6 | 0% | 67.1 | 0% | 58.5 | 19.2 | 0% |
| *MoD layer:* | | | | | | | | | | | |
| All layers | 45.9 | 38.2% | 42.6 | 33.7% | 25.9 | 32.8% | 33.8 | 34.1% | 37.1 | 12.3 | 34.7% |
| 1 MoD per 2 layers | 57.8 | 19.1% | 52.3 | 16.5% | 26.9 | 16.6% | 54.0 | 17.9% | 47.8 | 16.1 | 17.5% |
| 2 MoDs per 3 layers | 38.1 | 26.8% | 46.5 | 24.6% | 24.3 | 24.4% | 42.1 | 24.9% | 37.8 | 15.9 | 25.2% |
| ARank-based deployment | 63.7 | 40.7% | 68.5 | 35.9% | 35.6 | 36.8% | 65.3 | 38.2% | 58.3 | 12.6 | 37.9% |
| *Masked token:* | | | | | | | | | | | |
| None | 63.2 | 52.0% | 66.8 | 46.9% | 33.9 | 47.0% | 64.7 | 49.8% | 57.2 | 10.7 | 48.9% |
| Q | 63.7 | 40.7% | 68.5 | 35.9% | 35.6 | 36.8% | 65.3 | 38.2% | 58.3 | 12.6 | 37.9% |
| Q + A | 62.8 | 38.8% | 68.6 | 30.5% | 34.7 | 35.4% | 62.0 | 37.2% | 57.0 | 13.0 | 35.5% |
| *Shared router:* | | | | | | | | | | | |
| Not Share | 60.6 | 55.8% | 64.5 | 48.2% | 32.1 | 48.9% | 58.4 | 52.9% | 53.9 | 10.3 | 51.5% |
| Share | 63.1 | 60.3% | 67.9 | 56.9% | 34.7 | 56.6% | 64.9 | 57.1% | 57.6 | 9.3 | 57.7% |
| *Routing ratio:* | | | | | | | | | | | |
| 17% | 63.6 | 18.9% | 68.9 | 15.5% | 34.7 | 14.7% | 66.1 | 16.5% | 58.3 | 16.3 | 16.4% |
| 34% | 63.7 | 40.7% | 68.5 | 35.9% | 35.6 | 36.8% | 65.3 | 38.2% | 58.3 | 12.6 | 37.9% |
| 51% | 63.1 | 60.3% | 67.9 | 56.9% | 34.7 | 56.6% | 64.9 | 57.1% | 57.6 | 9.3 | 57.7% |
| 68% | 59.1 | 77.8% | 70.1 | 73.5% | 33.7 | 71.8% | 58.4 | 74.1% | 55.3 | 6.5 | 74.3% |

Table 2: **Ablation study of $\gamma$-MoD on LLaVA-HR.** "*Param*", "*Acc.*" and "*Skip*" indicate the parameter, accuracy, and skip ratio, respectively.

| Methods | Param | GQA | | SQA | | MMMU | | TextVQA | | Average | | |
|---|---|---|---|---|---|---|---|---|---|---|---|---|
| | | Acc. | Skip | Acc. | Skip | Acc. | Skip | Acc. | Skip | Acc. | TFlops | Skip |
| LLaVA-HR (Luo et al., 2024) | 7.4B | 64.2 | 0% | 67.9 | 0% | 34.6 | 0% | 67.1 | 0% | 58.5 | 19.2 | 0% |
| + Default MoD (Raposo et al., 2024) | 7.4B | 45.9 | 38.2% | 42.6 | 33.7% | 25.9 | 32.8% | 33.8 | 34.1% | 37.1 | 12.3 | 34.7% |
| + ARank-based deployment (ours) | 7.4B | 63.2 | 52.0% | 66.8 | 46.9% | 33.9 | 47.0% | 64.7 | 49.8% | 57.2 | 10.7 | 48.9% |
| + Masked routing learning (ours) | 7.4B | 63.1 | 60.3% | 67.9 | 56.9% | 34.7 | 56.6% | 64.9 | 57.1% | 57.6 | 9.3 | 57.7% |

report all the results in their default settings. In addition, we evaluate $\gamma$-MoD on six image question answering benchmarks: VQAv2 (Goyal et al., 2017), VizWiz (Gurari et al., 2018), TextVQA (Singh et al., 2019), SQA (Lu et al., 2022), GQA (Hudson & Manning, 2019) and SEED (Ge et al., 2023). We report all the results in their default settings. For MME, we report the perception score.

## 5.2 IMPLEMENTATION DETAILS

For all models, pre-training is conducted on LCS-558K dataset (Liu et al., 2024b), which includes high-quality 558k image-text pairs. For instruction tuning, we follow LLaVA-1.5 (Liu et al., 2024b) to use 665k vision-language instruction data. To deploy $\gamma$-MoD to MLLMs, ARank is calculated to identify redundant layers after the pre-training stage. For all models, the fourth largest ARank value is used as the threshold for converting dense layers to MoD ones. During instruction tuning, the coefficient for the routing loss is set to 0.01. The remaining settings are kept the same with LLaVA-HR (Luo et al., 2024) and LLaVA (Liu et al., 2024b), including learning rate, training epochs, optimizer and datasets, *etc*.

## 5.3 EXPERIMENTAL RESULTS

### 5.3.1 QUANTITATIVE ANALYSIS

**Comparison with different MoD configurations.** In Tab. 1, we first compare different settings of MoD on LLaVA-HR (Luo et al., 2024). From this table, the first observation is that directly converting all layers to MoD ones leads to worse results, *e.g.*, 33.8% on TextVQA. Besides, although the hand-craft strategy performs much better, its performance declines are still obvious, *e.g.*, -10.7% of 1 MoD per 2 layers on average. These results confirm the challenges of adopting MoDs to MLLMs.

Table 3: **Results of $\gamma$-MoD on different MLLM architectures and model scales.** $\gamma$-MoD-0.3 and $\gamma$-MoD-0.5 denote the routing ratio of 30% and 50%, respectively.

| Methods | Param | GQA | | SQA | | MMMU | | TextVQA | | Average | | |
|---|---|---|---|---|---|---|---|---|---|---|---|---|
| | | Acc. | Skip | Acc. | Skip | Acc. | Skip | Acc. | Skip | Acc. | TFlops | Skip |
| *MLLM architecture:* | | | | | | | | | | | | |
| LLaVA | 7B | 62.0 | 0% | 66.8 | 0% | 34.3 | 0% | 58.2 | 0% | 55.3 | 10.7 | 0% |
| +$\gamma$-MoD-0.3 | 7B | 61.1 | 34.1% | 64.7 | 29.4% | 35.4 | 29.8% | 56.3 | 30.7% | 54.4 | 7.7 | 31.0% |
| +$\gamma$-MoD-0.5 | 7B | 41.4 | 60.9% | 62.3 | 54.8% | 31.0 | 53.6% | 42.9 | 56.2% | 44.4 | 5.3 | 56.4% |
| LLaVA-HR | 7B | 64.2 | 0% | 67.9 | 0% | 34.6 | 0% | 67.1 | 0% | 58.5 | 19.2 | 0% |
| +$\gamma$-MoD-0.3 | 7B | 63.7 | 40.7% | 68.5 | 35.9% | 35.6 | 36.8% | 65.3 | 38.2% | 58.3 | 12.6 | 37.9% |
| +$\gamma$-MoD-0.5 | 7B | 63.1 | 60.3% | 67.9 | 56.9% | 34.7 | 56.6% | 64.9 | 57.1% | 57.6 | 9.3 | 57.7% |
| Mini-Gemini-HD | 7B | 62.9 | 0% | 69.6 | 0% | 36.8 | 0% | 66.5 | 0% | 59.0 | 60.2 | 0% |
| +$\gamma$-MoD-0.3 | 7B | 62.1 | 37.1% | 69.0 | 34.6% | 34.1 | 36.4% | 66.4 | 36.6% | 57.9 | 39.4 | 36.2% |
| +$\gamma$-MoD-0.5 | 7B | 62.2 | 59.2% | 70.4 | 56.8% | 33.9 | 58.6% | 67.0 | 57.7% | 58.4 | 27.8 | 58.1% |
| *Model scales:* | | | | | | | | | | | | |
| LLaVA-HR | 7B | 64.2 | 0% | 67.9 | 0% | 34.6 | 0% | 67.1 | 0% | 58.5 | 19.2 | 0% |
| +$\gamma$-MoD-0.3 | 7B | 63.7 | 40.7% | 68.5 | 35.9% | 35.6 | 36.8% | 65.3 | 38.2% | 58.3 | 12.6 | 37.9% |
| +$\gamma$-MoD-0.5 | 7B | 63.1 | 60.3% | 67.9 | 56.9% | 34.7 | 56.6% | 64.9 | 57.1% | 57.6 | 9.3 | 57.7% |
| LLaVA-HR | 13B | 64.8 | 0% | 68.1 | 0% | 36.7 | 0% | 68.1 | 0% | 59.4 | 37.1 | 0% |
| +$\gamma$-MoD-0.3 | 13B | 64.5 | 38.1% | 70.5 | 33.1% | 37.8 | 32.5% | 67.0 | 36.0% | 60.0 | 25.1 | 34.9% |
| +$\gamma$-MoD-0.5 | 13B | 64.8 | 58.8% | 69.5 | 52.2% | 35.8 | 53.8% | 66.8 | 55.4% | 59.2 | 18.4 | 55.1% |

Table 4: **Training and inference efficiency of $\gamma$-MoD on LLaVA-HR.** The inference efficiency is tested on an NVIDIA A100 GPU, which is the average value of GQA, SQA, MMMU, and TextVQA.

| Methods | Training Time ↓ | Inference Throughput ↑ | Inference Memory ↓ | Inference TFlops ↓ | Avg. Acc. ↑ |
|---|---|---|---|---|---|
| LLaVA-HR | 20.7 h | 4.7 samples/s | 19 G | 19.2 | 58.5 |
| +$\gamma$-MoD-0.3 | 15.4 h | 5.9 samples/s | 15 G | 12.6 | 58.3 |
| +$\gamma$-MoD-0.5 | 14.3 h | 7.2 samples/s | 14 G | 9.3 | 57.6 |
| Gains | **-31.0%** | **+53.2%** | **-26.3%** | **-51.6%** | **-0.9%** |

However, after employing our ARank-based strategy, the efficiency of LLaVA-HR is greatly increased while the performance is well maintained.

In Tab. 1, we also validate different micro-designs for deploying MoD on MLLM, including the masked routing learning, the shared router and the routing ratio. From these comparisons, we first see that the masked learning strategy is much beneficial to the optimization of $\gamma$-MoD, providing up to +1.7% gains on SQA. In addition, we also find that the router sharing strategy plays a significant role in $\gamma$-MoD. After removing this strategy, model performance will obviously drop on TextVQA by -6.5%. For routing threshold, we observe that the adaptive thresholds perform better while the default one is more efficient. Finally, we validate the impact of different routing ratio on LLaVA-HR. From results we can see that model performance can be retained under relatively small routing ratios, *i.e.,* 17% and 34%. When routing ratio is increased to 51%, model performance drops slightly from 58.3% to 57.6% on average. However, the benefit of efficiency is still notable, *i.e.,* -51.5% Flops.

**Ablation studies.** To validate contributions of each design in $\gamma$-MoD, we conduct ablation study in Tab. 2. From this table, we can see that the default MoD will cause obvious performance degeneration, resulting up to -25.3% on SQA. In stark contrast, with our ARank-based deployment, the average performance of LLaVA-HR is improved from 37.1% to 57.6%, and the computational sparsity also boosts from 34.7% to 48.9%. Such comparison confirms that not all layers can be converted to MoD layers, and ARank is critical to identify the redundant ones. In addition, the use of masked routing learning can further benefit the model training, providing +0.8% on MMMU and +0.2% on TextVQA, respectively. These results further confirm the effectiveness of $\gamma$-MoD.

### 5.3.2 COMPARISON WITH EXISTING MLLMS

**Generalizations of $\gamma$-MoD on different MLLMs.** In Tab. 3, we also evaluate the generalization capability of $\gamma$-MoD across different MLLM architectures and model scales. In particular, $\gamma$-MoD with 30% routing ratio demonstrates great trade-off between performance and efficiency on LLaVA. When

Table 5: **Comparison with quantization and pruning methods.** "*Speed*", "*Prefilling*" and "*Next-token*" indicate the throughput (samples/s), prefilling time (seconds) and next-token time (seconds), respectively. For MMMU, models predict one option without the need of the next-token time.

| Methods | MMMU | | | MM-Vet | | | |
|---|---|---|---|---|---|---|---|
| | Acc. | Speed | Prefilling | Acc. | Speed | Prefilling | Next-token |
| LLaVA-v1.5-7B (Liu et al., 2024b) | 34.3 | 9.1 | 0.11 | 30.5 | 0.53 | 0.20 | 1.8 |
| + AWQ-4bit (Lin et al., 2024b) | 34.8 | 11.1 | 0.09 | 26.5 | 0.57 | 0.16 | 1.6 |
| + FastV(K=2,R=50%) (Chen et al., 2025) | 33.9 | 11.6 | 0.09 | 28.8 | 0.68 | 0.17 | 1.3 |
| + $\gamma$-MoD-0.3 | 35.4 | 12.5 | 0.08 | 29.1 | 0.76 | 0.19 | 1.1 |

Table 6: **Comparison with existing dense and sparse MLLMs on 9 benchmarks.** Speed is the average samples per second of GQA, SQA, MMMU, and TextVQA.

| Methods | Param. | Image Question Answering | | | | Benchmark Toolkit | | | | | Speed |
|---|---|---|---|---|---|---|---|---|---|---|---|
| | | TextVQA | VQA$^{v2}$ | GQA | SQA$^{I}$ | POPE | MME | MMB | MMMU | MM-Vet | |
| *Dense Model:* | | | | | | | | | | | |
| I-80B (Laurençon et al., 2024) | 65B | - | 60.0 | 45.2 | - | - | - | 54.5 | - | - | - |
| InstructBLIP (Dai et al., 2023) | 14B | 50.7 | - | 49.5 | 63.1 | 78.9 | 1212.8 | - | - | 25.6 | - |
| VILA (Lin et al., 2024c) | 7B | 64.4 | 79.9 | 62.3 | 68.2 | 85.5 | 1533.0 | **68.9** | - | **34.9** | - |
| Qwen-VL (Bai et al., 2023b) | 10B | 63.8 | 78.8 | 59.3 | 67.1 | - | 1487.6 | 38.2 | - | - | 4.6 |
| LLaVA-1.5 (Liu et al., 2024b) | 7B | 58.2 | 78.5 | 62.0 | 66.8 | 85.9 | 1510.7 | 64.3 | 34.3 | 30.5 | 8.1 |
| LLaVA-HR (Luo et al., 2024) | 7B | 67.1 | 81.9 | 64.2 | 67.9 | 87.6 | **1554.9** | 66.8 | 35.2 | 31.2 | 4.7 |
| LLaVA-HR (Luo et al., 2024) | 13B | **68.1** | 82.3 | **64.8** | 68.1 | **87.8** | 1540.9 | 64.5 | **36.3** | 34.8 | 3.1 |
| *Sparse Model:* | | | | | | | | | | | |
| MoE-LLaVA (Lin et al., 2024a) | 3B | 50.1 | 76.7 | 60.3 | 62.6 | 85.7 | 1318.2 | 60.2 | - | 26.9 | 8.5 |
| MoE-LLaVA (Lin et al., 2024a) | 5B | 51.4 | 77.6 | 61.4 | 68.5 | 86.3 | 1423.0 | 65.2 | - | 34.3 | 5.6 |
| $\gamma$-MoD-LLaVA(ours) | 7B | 56.3 | 77.6 | 61.1 | 64.7 | 86.0 | 1342.1 | 59.4 | 35.4 | 29.8 | 10.3 |
| $\gamma$-MoD-LLaVA-HR(ours) | 7B | 64.9 | 80.6 | 63.1 | 67.9 | 87.3 | 1516.0 | 63.4 | 34.7 | 31.5 | 7.2 |
| $\gamma$-MoD-LLaVA-HR(ours) | 13B | 66.8 | 82.0 | **64.8** | **69.5** | 86.7 | 1515.4 | 65.2 | 35.8 | 34.0 | 4.8 |

the routing ratio increases to 51%, the performance of LLaVA decreases significantly, suggesting its relatively low tolerance to high routing ratio. For LLaVA-HR, the $\gamma$-MoD-0.3 configuration maintains high accuracy 63.7% on GQA and 65.3% on TextVQA while reducing TFlops by 34% and skipping 37.9% of tokens. When the routing ratio increases to 51%, the token skip rate improves to 57.7%, though a slight drop in accuracy is observed *e.g.,* -0.6% on GQA. These comparisons also reflect that high-resolution MLLMs often have a higher token redundancy than low-resolution ones. Similar observations can also be witnessed on Mini-Gemini-HD Li et al. (2024b). When scaling to larger models, such as the LLaVA-HR-13B, our method continues to perform strongly. The $\gamma$-MoD-0.3 configuration yields a 38.1% skip rate and 25.1 TFlops with minimal accuracy loss, suggesting that larger models are better suited to handle higher skip rates while maintaining performance. Even increasing the routing ratio to 51% the competitive accuracy is still maintained, *e.g.,* 64.8% on GQA and 66.8% on TextVQA.

**Efficiency analysis.** In Tab. 4, we compare the training and inference efficiency of $\gamma$-MoD on LLaVA-HR. From these results, we observe comprehensive advantages of $\gamma$-MoD in terms of training and inference inference. In particular, $\gamma$-MoD-0.3 already achieves an obvious improvement in efficiency, *i.e.,* -26% training time and -35% TFlops. However, the performance drops of $\gamma$-MoD-0.3 can be almost ignorable, *i.e.,* -0.2% average accuracy. When increasing the routing ratio to 50% tokens, the inference throughput of $\gamma$-MoD-0.5 further improves by up to +53.2%. Despite the significant efficiency gains, the performance drop of $\gamma$-MoD is still acceptable, *i.e.,* -1.5% average accuracy. These results well validate the obvious benefits of $\gamma$-MoD in efficiency.

**Comparison with existing methods.** In Tab. 6, we compare MLLMs deployed by $\gamma$-MoD with both dense and sparse models on 9 benchmarks. From it we can see $\gamma$-MoD can maintain the competitive performance on all benchmarks, while achieving significant efficiency gains on LLaVA and LLaVA-HR. Specifically, $\gamma$-MoD-LLaVA-HR (13B) can reach similar inference speed as LLaVA-HR (7B) while outperforming the latter on multiple benchmarks, *e.g.,* +3.0% on MMVet. In addition, compared to existing sparse models, *i.e.,* MoE-LLaVA (Lin et al., 2024a), our approaches also achieve better trade-off between performance and efficiency. In particular, $\gamma$-MoD-LLaVA-HR (7B) outperforms MoE-LLaVA (5B) on 5 of 8 benchmarks, *e.g.,* + 93 scores on MME, while still maintaining better efficiency, *i.e.,* +28% gains on inference speed. It is worth noting that although the parameter scale of MoE-LLaVA is smaller, its routing calculation often leads to higher latency.

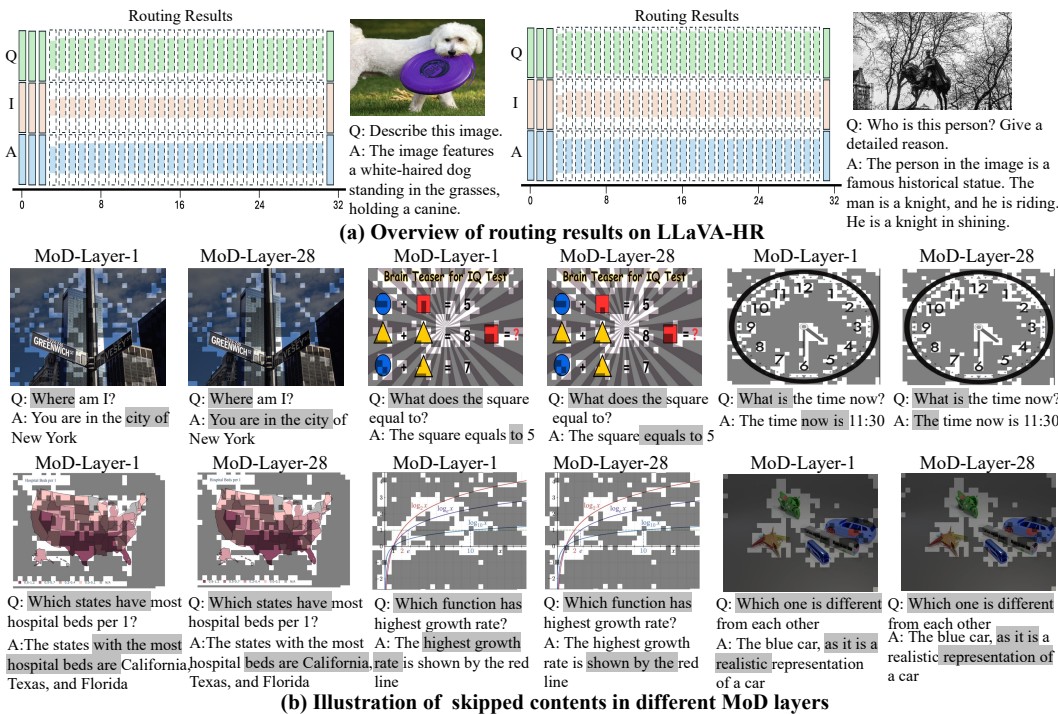

Figure 4: **Visualization of routing results for different MoD layers.** "Q", "I" and "A" denote the question, image and response, respectively. The skipped tokens in sub-figure (b) are colored in gray.

In Tab. 5, we also compare $\gamma-$MoD with common inference-time acceleration methods Chen et al. (2025); Lin et al. (2024b). Compared to these methods, $\gamma-$MoD can better maintain the model performance on MMMU and MMVet. In terms of efficiency, $\gamma-$MoD shows greater advantages in accelerating the next-token generation for MLLMs, providing up to +31% speedups on MMVet. Overall, these comparisons further confirm the effectiveness and efficiency of $\gamma$-MoD.

### 5.3.3 QUALITATIVE ANALYSIS

In Fig. 4, we visualize the routing ratio and the skipped content in both images and the corresponding conversations. The first observation from Fig. 4.(a) is that question, image, and response tokens are routed in a consistent pattern: question tokens are mostly kept, while image tokens are the most redundant, and thus routed the most. In Fig. 4.(b), we visualize the skipped content on images and texts. The gray portions of the images represent tokens that are skipped by the router, indicating that many regions in the images, such as background pixels, are redundant and do not provide critical information for understanding. Routing out these tokens allows the model to focus more on the white portions, which highlight the image regions or text parts that the model pays closer attention to. For example, in the middle of the first row with the IQ test example, the model can concentrate and spending more computations on the arithmetic and geometric aspects of the image.

## 6 CONCLUSION

In this paper, we aim to overcome the efficiency problem in multimodal large language models (MLLMs) from the perspective of "activated token". In particular, we present $\gamma$-MoD, a novel mixture-of-depth adaptation strategy for computationally efficient MLLM. In $\gamma$-MoD, an innovative metric is introduced to identify the redundant layers for MoD deployment, namely *rank of attention maps* (ARank). Moreover, $\gamma$-MoD also maximizes its benefit to MLLMs via two designs called *shared vision-language router* and *masked routing learning*. With these novel designs, $\gamma$-MoD can obviously reduce computational costs of existing MLLMs while maintaining their performance. Extensive experiments on 9 multimodal benchmarks validate the efficiency and effectiveness. Besides, the great generalization ability of $\gamma$-MoD is also validated across different MLLMs.

**Acknowledgments.** This work was supported by the National Natural Science Foundation of China (No. 623B2088) and the China Postdoctoral Science Foundation (No. 2024M761548).

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
