# OpenReview forum: "$\gamma-$MoD: Exploring Mixture-of-Depth Adaptation for Multimodal Large Language Models"
_ICLR.cc/2025/Conference — ICLR 2025 Poster_

### Official Review · Reviewer_wmRq · 2024-10-20

**Soundness:** 3
**Presentation:** 3
**Contribution:** 3
**Rating:** 8
**Confidence:** 4

**Summary:**

This paper proposes γ-MoD, a novel mixture-of-depth (MoD) adaptation strategy for existing multimodal large language models (MLLMs). The goal is to maximize computational sparsity while maintaining competitive performance. γ-MoD achieves this by identifying redundant layers using the rank of attention maps (ARank) metric and replacing them with MoD layers. The Key contributions of this paper include: **1.** γ-MoD Framework: A novel MoD framework for sparse computation of existing MLLMs, seamlessly converting most dense layers to sparse MoD layers. **2.** ARank Metric: An metric to measure layer redundancy, guiding the selection of layers for MoD adaptation. **3.** Shared Vision-Language Router and Masked Routing Learning: Designs to maximize the benefits of MoDs in MLLMs, achieving up to 51.6% computational sparsity with minimal performance sacrifice.

**Strengths:**

1. While MoD has been explored in LLMs, this is the first work to adapt them to MLLMs, which face unique challenges due to the combination of visual and textual modalities.
2. This paper proposes a metric, rank of attention maps (ARank), to identify redundant layers in MLLMs, addressing the lack of guidance in existing MoD approaches.
3. The paper introduces two designs to maximize the benefits of MoD for MLLMs, The Masked Routing Learning prevents critical tokens, such as question subjects, from being skipped during training, preserving the model’s ability to generate accurate answers.
4. This paper provides comprehensive theoretical analysis, empirical validations. And it is well-structured and clearly explains the motivations, methodology, and experimental results.

**Weaknesses:**

1. The metric ARank seems to be a repurposed version of HRank [1] specifically for attention maps. This connection should be discussed in detail.
2. The ablation study of the routing mechanisms is not enough. This paper only explores a binary routing mechanism (skip or not skip). Exploring more sophisticated routing strategies, such as weighted routing or adaptive routing thresholds and so on, could make the article more comprehensive.
3. While the paper compares γ-MoD with dense and MoE-based MLLMs, it lacks a comparison with other sparsity techniques like pruning or quantization.
4. The author should consider rearranging Table 5 or highlighting the optimal results, as the current version is somewhat challenging to compare the results of different methods

[1] Mingbao Lin, Rongrong Ji, Yan Wang, Yichen Zhang, Baochang Zhang, Yonghong Tian, and Ling Shao. Hrank: Filter pruning using high-rank feature map. In Proceedings of the IEEE/CVF conference on computer vision and pattern recognition, pp. 1529–1538, 2020.

**Questions:**

1. In this paper, the binary mask Mq assigned the question tokens to 0. Therefore, during the training process, the value of Mq should be fixed. But for different tasks or datasets, the location of the question tokens may be different. How did the author determine and design Mq on different tasks or datasets?
2. I would like to know how the "All layer" in Table 1 is specifically implemented.
3. How sensitive is ARank to the choice of hyperparameters, such as the batch used for estimation?
4. What are the criteria for determining the threshold for converting dense layers to MoD layers based on ARank values? Is there a more principled way to set this threshold?

**Details Of Ethics Concerns:**

No.

---

> ### Author Response · Authors · 2024-11-22
>
> ---
>
> > **Comment 1:  The metric ARank seems to be a repurposed version of HRank [1] specifically for attention maps. This connection should be discussed in detail.**
>
> **Response**: Thanks for your feedback.  ARank is inspired by HRank [A] but still quit different in **theoretical and practical contributions**. In particular, HRank aims to use the rank of feature maps to estimate the weight redundancy for network pruning.  In contrast, we are the first to validate that the layer redundancy  can also be estimated by the rank of their attention maps, thus guiding the deployment of MoD.   Compared to HRank, we have extended the theory and the practical use of the ''rank''.  Based on your advice, we will add these discussions in our new revision.
>
> [A] Hrank: Filter pruning using high-rank feature map, CVPR 2020
>
> ---
>
>
> > **Comment 2: The ablation study of the routing mechanisms is not enough. This paper only explores a binary routing mechanism (skip or not skip). Exploring more sophisticated routing strategies, such as weighted routing or adaptive routing thresholds and so on, could make the article more comprehensive.**
>
>
>
> **Response**: Thanks for this constructive advice.  In fact, binary routing mechanism could best meet the autoregressive property of MLLMs, where the latter tokens cannot see the former ones.  Therefore, it's pretty difficult to weighted aggregate several tokens into one, which may break the autoregressive property.
>
>
>
> However, we do agree that the adaptive routing thresholds should be a promising approach. Driven by this, we consider two adaptive metrics based on the sequence length and the ARank, defined by $\delta_s*(1+ \log(\frac{seqlen}{avgseqlen}))$  and $\delta_s*(1+ \log(\frac{ARank}{avgARank}))$ , respectively.   Experimental results show the adaptive metrics can slightly benefit the model performance on multiple benchmarks.  As you suggested, these  results would be added in our final version.
>
>
>
> | Method           | SQA ACC | SQA Skip | GQA ACC | GQA Skip | MMMU ACC | MMMU Skip | TextVQA ACC | TextVQA Skip |
> | ---------------- | ------- | -------- | ------- | -------- | -------- | --------- | ----------- | ------------ |
> | $\gamma$-MoD-0.5 | 67.9    | 56.9%    | 63.1    | 60.3%    | 34.7     | 56.6%     | 64.9        | 57.1%        |
> | + Seqlen        | 67.7    | 55.8%    | 62.9    | 56.3%    | 35.4     | 55.3%     | 64.7        | 57.4%        |
> | + ARank          | 68.1    | 54.2%    | 63.2    | 57.9%    | 34.3     | 53.8%     | 64.6        | 55.9%        |
>
> ---
>
>
> > **Comment 3:  While the paper compares γ-MoD with dense and MoE-based MLLMs, it lacks a comparison with other sparsity techniques like pruning or quantization.**
>
>
>
> **Response**:   We fully understand your concerns and have compared $\gamma$-MoD with token pruning  method (Fast-V [B]) and quantization method (AWQ [C]) in the table below.   In this table, we test the speed of 4-bit model (awq-4bit) using the lmdeploy toolkit [D], while others are evaluated using Pytorch.  From these results, we still observe the advantage of  $\gamma$-MoD in performance and efficiency over pruning and quantization, especially in reducing the next-token time.
>
>
>
> We believe that these comparisons is highly beneficial to our paper, and will update them in our final version.
>
> | Model                | MMMU Accuracy (%) | Speed (samples/s) | Prefilling Time (s)  | MM-Vet Accuracy (%) | Speed (samples/s) | Prefilling Time (s) | Next-token Time (s) |
> | -------------------- | ----------------- | ----------------- | ------------------- | ------------------- | ----------------- | ------------------- | ------------------- |
> | LLaVA-v1.5-7b        | 34.3              | 9.1              | 0.11                  | 30.5                | 0.53              | 0.20                | 1.8                 |
> | +awq-4bit            | 34.8              | 11.1              | 0.09                    | 26.5                | 0.57              | 0.16                | 1.6                 |
> | + FastV (K=2, R=50%) | 33.9              | 11.6              | 0.09          | 28.8                | 0.68              | 0.17                | 1.3                 |
> | +gamma-MoD-0.34      | 35.4              | 12.5               | 0.08         | 29.1                | 0.76              | 0.19                | 1.1                 |
>
> [B] An Image is Worth 1/2 Tokens After Layer 2: Plug-and-Play Inference Acceleration for Large Vision-Language Models, ECCV24
>
>
>
> [C] AWQ: Activation-aware Weight Quantization for LLM Compression and Acceleration, MLSys 2024
>
>
>
> [D] LMDeploy: A Toolkit for Compressing, Deploying, and Serving LLM
>
> ---

---

> ### Author Response · Authors · 2024-11-22
>
> ---
>
>
> > **Comment 4:  The author should consider rearranging Table 5 or highlighting the optimal results, as the current version is somewhat challenging to compare the results of different methods.**
>
>
>
> **Response**:  Thanks for your kind reminder.   We will carefully adjust the layout and format of Table 5 to improve its readability.
>
>
>
> ---
>
>
>
> > **Comment 5: In this paper, the binary mask Mq assigned the question tokens to 0. Therefore, during the training process, the value of Mq should be fixed. But for different tasks or datasets, the location of the question tokens may be different. How did the author determine and design Mq on different tasks or datasets?**
>
>
>
> **Response**: In practice, the positions and lengths of questions in a batch are different, so we assign corresponding binary masks to them. Therefore, $M_q$ will be changed based on the input sample.
>
> ---
>
>
>
> > **Comment 6:  I would like to know how the "All layer" in Table 1 is specifically implemented.**
>
>
>
> **Response**: "All layer" means that all dense layers are replaced with the MoD ones.
>
> ---
>
>
>
> > **Comment 7:  How sensitive is ARank to the choice of hyperparameters, such as the batch used for estimation?**
>
>
>
> **Response**: Actually, ARank is highly stable to  the change of tasks and samples, and a small batch of samples (50 in our paper) can already accurately estimate its value.  In Fig 3, we visualize the impact of tasks and batch size on ARank, which may help you further understand the robustness of ARank.
>
> ---
>
>
>
> > **Comment  8:  What are the criteria for determining the threshold for converting dense layers to MoD layers based on ARank values? Is there a more principled way to set this threshold?**
>
>
>
> **Response**: In $\gamma$-MoD,  the threshold is still a hyperparameter that needs to be manually selected, and this setting in our paper is sufficient to validate the effectiveness of $\gamma$-MoD.  However, as you suggested in **Comment 2**, the adaptive threshold would be a better alternative,  which we will further explore in future work.
>
> ---

---

> > ### Comment · Reviewer_wmRq · 2024-11-24
> >
> > Thanks for the response from the authors. Most of my concerns are addressed, I have raised my score accordingly.

---

> > > ### Author Response · Authors · 2024-11-24
> > >
> > > Thanks for your encouraging decision! We would like to express our appreciation again for your invaluable suggestions and professional comments.

---

### Official Review · Reviewer_6GUt · 2024-10-23

**Soundness:** 4
**Presentation:** 3
**Contribution:** 3
**Rating:** 6
**Confidence:** 5

**Summary:**

This article introduces the Mixture-of-Depth (MoD) scheme into multimodal large language model (MLLM) architectures. By analyzing the rank of attention maps (ARank) of existing MLLMs, the authors identify a model-agnostic and data-agnostic pattern, revealing significant redundancy in certain layers of MLLMs. Building on this insight, the paper proposes the $\gamma$-MoD method, which (1) incorporates a shared vision-language router into existing MLLMs, transforming them into the MoD structure, and (2) employs masked routing learning to adapt MLLM architectures to the MoD scheme by refining the instruction tuning process used during MLLM training. Extensive experiments on multiple multimodal benchmarks and MLLM models demonstrate that $\gamma$-MoD effectively reduces both training and inference time, while largely preserving performance on benchmark tasks.

**Strengths:**

1. Originality: I appreciate the authors' effort in implementing the MoD scheme in MLLMs in a plug-and-play manner. In my view, the significance of this work lies in maintaining the pre-training stage unchanged, which contrasts with most approaches in the field of sparse LLMs/MLLMs (e.g., MoE structures) that focus on pre-training models from scratch. While there are some aspects that warrant further consideration (see weaknesses & questions), this approach could inspire future research into converting existing models into sparse structures in an off-the-shelf manner.
2. Clarity: This article is mostly well-written. The motivation is clearly articulated and the solutions towards solving the existing problems are logically sound.
3. Experiments: This article conducts extensive experiments across multiple MLLM structures and benchmark datasets to demonstrate the generalizability of $\gamma$-MoD. Their results strongly suggest that $\gamma$-MoD successfully reduces training and inference cost on MLLM models. Abundant analysis experiments also demonstrated that $\gamma$-MoD leads MLLMs to focus on informative tokens and skip the redundant ones.

**Weaknesses:**

1. Method designs: In the introduction (Line 87), this article suggests that $\gamma$-MoD is a *plug-and-play* method that can be integrated to current MLLMs. Existing MLLMs follow a 2-stage pre-training and instruction tuning training pipeline, and $\gamma$-MoD modifies the instruction tuning pipeline to convert LLaVA models in $\gamma$-MoD-LLaVA structure. In my opinion, this design is arguably *plug-and-play*, since open-source MLLM weights are mostly instruction tuned, so that $\gamma$-MoD cannot be directly applied on published MLLM checkpoints. To evaluate the validity of $\gamma$-MoD, the authors also pre-trains MLLMs on the LCS-558K dataset first. This hinders the applicability of $\gamma$-MoD, as training the $\gamma$-MoD model also requires conducting the complete pre-training pipeline.
2. Further generalizability of $\gamma$-MoD: The experiments in this article primarily focus on LLaVA structures, likely due to the availability of open-source pre-training and instruction tuning datasets, which are necessary for $\gamma$-MoD as it modifies the instruction tuning process. However, as a general method for introducing the MoD scheme to MLLM models, additional experiments on other MLLMs are needed to further demonstrate the generalizability of $\gamma$-MoD.
3. Some ambiguities in the methodology: Key information are missing from the article & Some statements are quite confusing. See Questions for more details.
4. Significance when compared to existing works: In the experiments section, this article finds that most image tokens can be skipped in MLLM inference. This discovery corresponds with [1], in which image tokens are simply pruned from off-the-shelf MLLMs without further training. experiment results in [1] indicates that this simple procedure reduces the FLOPs in MLLMs by over 50% without introducing performance drops, similar to that reported in $\gamma$-MoD. If so, pre-train MLLMs from scratch and modify the instruction tuning methods would be too costly for accelerating an MLLM.

[1] An Image is Worth 1/2 Tokens After Layer 2: Plug-and-Play Inference Acceleration for Large Vision-Language Models (ECCV 2024)

**Questions:**

Things that WILL affect my overall rating towards this article:

1. Regarding Weakness 1, since the authors consider the disadvantage of MoD to be demanding training from scratch (Line 49-50), the authors should demonstrate how to apply $\gamma$-MoD to existing published MLLM checkpoints, rather than pre-training one from scratch like in the experiment section.
2. Regarding Weakness 2, I would like to see more experiments on other MLLM models beside the LLaVA series. Considering the time of rebuttal and your reported time of training in Table 4, I would like to see this point resolved.
3. Regarding Weakness 3:
   - What is the applied value of the pre-defined sparse target in Equation 4? What is the relationship between the sparse target $\alpha$ and the "routing ratio" and the "skip ratio" in the experiment section?
   - Normally, the sparse target factor $\alpha$ in Equation 4 satisfies $\alpha\in[0, 1]$ , but $\alpha$ is actually the sum of multiple indicator functions with values $\\{0,1\\}$. The left side (value of the sum of indicator function) should be normalized.
   - In Equation 8, the authors put one-hot vectors $\hat{R}$ and $1-\hat{R}$ inside $\log$. This obviously doesn't seem right. Moreover, it is hard to understand what does the routing objective (Equation 8) do in $\gamma$-MoD training. The authors should provide a more detailed explanations of that learning objective.
4. Regarding Weakness 4: Since both works lead to inference acceleration of MLLMs, the authors should provide a detailed comparison with [1] and demonstrate the exclusive advantage of $\gamma$-MoD.

Things that WILL NOT affect my overall rating towards this article:

1. Figure 2 indicates that transformer layers with $\text{Rank}<110$ would be considered as redundant layers. The authors could provide a more detailed analysis on the rank distributions of tested models, instead of giving a figure without quantitative statistics like Figure 3.

[1] An Image is Worth 1/2 Tokens After Layer 2: Plug-and-Play Inference Acceleration for Large Vision-Language Models (ECCV 2024)

---

> ### Author Response · Authors · 2024-11-22
>
> ---
>
> > **Comment 1:Method designs: In the introduction (Line 87), this article suggests that γ-MoD is a *plug-and-play* method that can be integrated to current MLLMs. Existing MLLMs follow a 2-stage pre-training and instruction tuning training pipeline, and γ-MoD modifies the instruction tuning pipeline to convert LLaVA models in γ-MoD-LLaVA structure. In my opinion, this design is arguably *plug-and-play*, since open-source MLLM weights are mostly instruction tuned, so that γ-MoD cannot be directly applied on published MLLM checkpoints. To evaluate the validity of γ-MoD, the authors also pre-trains MLLMs on the LCS-558K dataset first. This hinders the applicability of γ-MoD, as training the γ-MoD model also requires conducting the complete pre-training pipeline.**
>
>
>
> **Response:**  Thanks for this comment.  We would like to clarify that  $\gamma$-MoD  is not a  *plug-and-play* inference module but a *plug-and-play* adaptation method that can be seamlessly integrated into the standard training process of MLLMs.   ***In fact,   $\gamma$-MoD  is not used for a well pre-trained MLLM, but serves for a common situation that we want to train a new MLLM in an efficient way.***  Considering for this,  the strengths of $\gamma$-MoD  focus on both training and inference efficiency, see Tab 4 (+31% and +53% speedups by   $\gamma$-MoD for training and inference, respectively).  However, we do agree that the description of plug-and-play is not rigorous enough, and we will carefully revise it in the final version.
>
>   ***Note that one detail we would like to kindly clarify that the pre-training process is not actually needed for  $\gamma$-MoD, since  $\gamma$-MoD is deployed in the instruction tuning stage. In practice, we use the published pre-training weight in our experiment, please see our anonymously released code: https://anonymous.4open.science/r/Gamma-MOD-code-AC17.***
>
>
>
> ---
>
> > **Comment 2:  Regarding Weakness 1, since the authors consider the disadvantage of MoD to be demanding training from scratch (Line 49-50), the authors should demonstrate how to apply γ-MoD to existing published MLLM checkpoints, rather than pre-training one from scratch like in the experiment section.**
>
>
>
> **Response**:  Thanks for this kind advice.  In Line 49-50, we stated that recent MoDs[A] require to pretrain **LLMs** from scratch, ***instead of MLLMs***.  Therefore, our motivation is that how to directly adapting MoDs to MLLMs without the expensive pre-training of a new LLM.  In this case, directly applying  γ-MoD to existing published MLLM checkpoints is indeed orthogonal to our contribution, as discussed in **Comment 1**.
>
>
>
> However, we fully respect your concerns and have attempted to apply  γ-MoD to published MLLM checkpoints in the table below (Direct  $\gamma$-MoD).  In particular, we modified the trainable router to an rule-based one, which skips tokens based on its attention scores. As expected,  the *Direct $\gamma$-MoD* can achieve promising results on some benchmarks like SQA, but is still inferior to our  *Default $\gamma$-MoD* on challenging benchmarks like TextVQA.  Based on your suggestion, we will further clarify our motivation and contribution in the final version.
>
>
>
> | Model                  | SQA ACC | SQA Skip | GQA ACC | GQA Skip | MMMU ACC | MMMU Skip | TextVQA ACC | TextVQA Skip |
> | ---------------------- | ------- | -------- | ------- | -------- | -------- | --------- | ----------- | ------------ |
> | LLaVA-HR               | 67.9    | 0%       | 64.2    | 0%       | 34.6     | 0%        | 67.1        | 0%           |
> | +Direct $\gamma$-MoD   | 66.1    | 50%      | 58.7    | 50%      | 33.7     | 50%       | 48.7        | 50%          |
> | + Default $\gamma$-MoD | 67.9    | 56%      | 63.1    | 60%      | 34.7     | 56%       | 64.9        | 57%          |
>
> [A] David Raposo, Sam Ritter, Blake Richards, Timothy Lillicrap, Peter Conway Humphreys, and Adam Santoro. Mixture-of-depths: Dynamically allocating compute in transformer-based language models. arXiv preprint arXiv:2404.02258, 2024.
>
>
>
> ---

---

> > ### Comment · Reviewer_6GUt · 2024-11-22
> >
> > Many thanks to the authors for the elaboration. The response has resolved some of my considerations on the applicability and usage of $\gamma$-MoD.
> >
> > However, I would like some further explanations on the following: *the pre-training process is not actually needed for
> > $\gamma$-MoD*. I have read the README.md in the codebase, and it DOES need a State-1, which is the pretraining stage.

---

> ### Author Response · Authors · 2024-11-22
>
> ---
> > **Comment 3: Further generalizability of γ-MoD: The experiments in this article primarily focus on LLaVA structures, likely due to the availability of open-source pre-training and instruction tuning datasets, which are necessary for γ-MoD as it modifies the instruction tuning process. However, as a general method for introducing the MoD scheme to MLLM models, additional experiments on other MLLMs are needed to further demonstrate the generalizability of γ-MoD.**
>
>
>
> **Response:**  We appreciate this constructive advice.  Following your suggestion, we adopted  γ-MoD to another popular MLLM termed Mini-Gemini [B] in the table below, and validated the generalizability of  γ-MoD  on this new MLLM architecture.
>
>
>
> Source codes and pre-trained weights are also anonymously released at: https://anonymous.4open.science/r/Gamma-MOD-code-AC17.  As the first MoD-based work for MLLMs, we sincerely hope that our open-source codes will benefit future works in the community.
>
> | Methods           | Param | GQA Acc. | GQA Skip | SQA Acc. | SQA Skip | MMMU Acc. | MMMU Skip | TextVQA Acc. | TextVQA Skip | Average Acc. | Average TFlops | Average Skip |
> | ----------------- | ----- | -------- | -------- | -------- | -------- | --------- | --------- | ------------ | ------------ | ------------ | -------------- | ------------ |
> | Mini-Gemini-HD    | 7B    | 62.9     | 0%       | 69.6     | 0%       | 36.8      | 0%        | 66.5         | 0%           | 59.0         | 60.2           | 0%           |
> | +$\gamma$-MoD-0.3 | 7B    | 62.1     | 37.1%    | 69.0     | 34.6%    | 34.1      | 36.4%     | 66.4         | 36.6%        | 57.9         | 39.4           | 36.2%        |
> | +$\gamma$-MoD-0.5 | 7B    | 62.2     | 59.2%    | 70.4     | 56.8%    | 33.9      | 58.6%     | 67.0         | 57.7%        | 58.4         | 27.8           | 58.1%        |
>
> [B] Mini-gemini: Mining the potential of multi-modality vision language models,  arXiv preprint arXiv:2403.18814, 2024.
>
>
>
> ---
>
> > **Comment 4: Significance when compared to existing works: In the experiments section, this article finds that most image tokens can be skipped in MLLM inference. This discovery corresponds with [C], in which image tokens are simply pruned from off-the-shelf MLLMs without further training. experiment results in [C] indicates that this simple procedure reduces the FLOPs in MLLMs by over 50% without introducing performance drops, similar to that reported in γ-MoD. If so, pre-train MLLMs from scratch and modify the instruction tuning methods would be too costly for accelerating an MLLM.**
>
>
>
> **Response:**   Thanks for this comment. We fully agree that [C] is a promising approach to directly reduce the computational cost of pre-trained MLLMs, and we will carefully discuss and compare it in the final version.  Nevertheless, as an inference-time acceleration method, [C] is quit orthogonal to the contribution of $\gamma-$MoD.  In terms of significance, we summarize the differences in two aspects:
>
>
> 1.**Efficiency during next-token generation:**  [C]  mainly aims to reduce the number of visual tokens in the pre-filling stage. ***In stark contrast,  $\gamma-$MoD can skip both visual and textual tokens in both the pre-filling and next-token generation stages***. As shown in the table below, the next-token generation time often accounts for a large proportion of the inference time, and $\gamma-$MoD can reduce the time of this stage by up to 58%.
>
> 2. **Training efficiency:** Instruction tuning is an an inevitable process for deploying a new MLLM.  ***As an adaptation approach, $\gamma-$MoD can further reduce the training time by up to 31% (see Tab 4), which is indeed practically useful (as recognized by Reviewer 1ATp).***
>
> Based on above discussions, we sincerely hope that you would reconsider the scope and contribution of our paper.
>
> | Model                   | Accuracy (MMVet) | Speed (samples/s) | Prefilling Time (s) | Next-Token Time (s) |
> | ----------------------- | ---------------- | ----------------- | ------------------- | ------------------- |
> | LLaVA-HR                | 31.2             | 0.38              | 0.224               | 2.6                 |
> | +FastV [B] (K=2, R=50%) | 27.2             | 0.57              | 0.154               | 2.1                 |
> | $\gamma$-MoD-0.34       | 33.3             | 0.76              | 0.171               | 1.3                 |
> | +$\gamma$-MoD-0.5       | 31.5             | 0.83              | 0.147               | 1.1                 |
>
> [C] An Image is Worth 1/2 Tokens After Layer 2: Plug-and-Play Inference Acceleration for Large Vision-Language Models (ECCV 2024)
>
> ---

---

> ### Author Response · Authors · 2024-11-22
>
> ---
>
> > **Comment 5: Some ambiguities in the methodology: Key information are missing from the article, some statements are quite confusing.**
>
>
> **Question 1:** What is the applied value of the pre-defined sparse target in Equation 4? What is the relationship between the sparse target α and the "routing ratio" and the "skip ratio" in the experiment section?
>
> **Response**: The pre-defined sparse target $\alpha$ is the routing ratio that the router is required to approximate during training, e.g., 0.5 for $\gamma-$MoD-0.5.  During inference, the actual routing ratio may be slightly different to the  training target  $\alpha$ .
>
>
>
> **Question 2:** Normally, the sparse target factor α in Equation 4 satisfies α∈[0,1] , but α is actually the sum of multiple indicator functions with values {0,1}. The left side (value of the sum of indicator function) should be normalized.
>
>
>
> **Response**: Yes, α should satisfy $\alpha \in$ [0,1]. We apologize for the lack of normalization in Equation 4 and will carefully revise it in the new version.
>
>
>
> **Question 3:** In Equation 8, the authors put one-hot vectors R^ and 1−R^ inside log. This obviously doesn't seem right. Moreover, it is hard to understand what does the routing objective (Equation 8) do in γ-MoD training. The authors should provide a more detailed explanations of that learning objective.
>
>
>
> **Response**: We would like to thank you again for these careful reviews. Equation 8 defines a binary cross-entropy loss for routing learning,  which encourages that the router can learn to discard unimportant tokens based on a pre-defined  sparse target $\alpha$. Our routing objective is equivalent to that of the original MoD[B]. Based on your feedback, we will carefully revise the description for it to avoid misleading.
>
>
>
> ---

---

> > ### Comment · Reviewer_6GUt · 2024-11-22
> >
> > Also, regarding Equation 8, I totally understand that it encourages the router to discard unimportant tokens. I have two questions on this:
> >
> > 1. How does the sparsity target $\alpha$ affect the objective in Equation 8?
> > 2. Why put $\hat{R}$ in side the $\log$ while the $R(x)\cdot M_q$ outside? If you want to apply the cross-entropy loss, you should do the opposite: put $R(x)\cdot M_q$ inside the $\log$ and the true label $\hat{R}$ outside.

---

> ### Author Response · Authors · 2024-11-22
>
> Thank you for your prompt reply! As pointed out in the paper, our $\gamma-$MoD is used for the instruction fine-tuning stage, so the pre-trained weights of the first stage can be directly reused in the publicly released stages (e.g. https://huggingface.co/YanweiLi/MGM-Pretrain). This means that we can start deploying $\gamma-$MoD based on the existing publicly available first stage pre-trained weights, thus skiping the pre-training stage. Actually, we also adopt this way in our experiments.
>
>
> Note that we miss these details in our paper, so feel sorry for  any misleading. We will carefully update these details in final version. We hope these further explanations can resolve your concerns and look forward to more discussions.

---

> > ### Comment · Reviewer_6GUt · 2024-11-22
> >
> > Thanks for the further explanations from the authors. I am looking forward to the revision of the article and will adjust the overall ratings based on that.

---

> ### Author Response · Authors · 2024-11-22
>
> 1. $\alpha$ actually defines the number of positive labels used to calculate the cross entropy loss. For example, if we have a sequence length of 100 and $\alpha$ is 0.5, there will be 50 labels assigned to 1 for routing learning.
> 2. We double-check the equoation, and confirm that you are correct. Sorry for the obvious typos. We'll double-check it one more time.

---

> ### Author Response · Authors · 2024-11-23
>
> Dear Reviewer 6GUt:
>
> We highly appreciate your recognition to our response, and your insightful reviews have greatly contributed to improving the clarity and overall quality of our work.  Now we have updated the revised paper based on your and other reviewers' concerns. The main modifications include:
> 1. Detailed discussions and comparisons with FastV in the related work (Line 148-152) and Tab 5. (Reviewer 6GUt)
> 2. Generilization results on Mini-Gemini in Tab 3. (Reviewer 6GUt)
> 3. Clarifications about the pre-training stage in Line 340-346. (Reviewer 6GUt)
> 4. Clarifications about the plug-and-play adaptation module in Line 340-346. (Reviewer 6GUt)
> 5. Results of adaptive thresholds in Tab 1. (Reviewer wmRq)
> 6. Discussions of differences with HRanks in  Line 297-300. (Reviewer wmRq)
> 7. Comparison with quantization and pruning methods in Tab 5. (Reviewer wmRq)
> 8.  Other details: typos like mistake equoations and highlights of the best value in the table, etc. (Reviewer 6GUt , wmRq)
> 9. Clarifications for the robustness of ARank in Line 269-304.  (Reviewer 1ATp)
>
> If issues are well addressed in our new revision, we sincerely hope you could  kindly raise the score.
>
> Best,
>
> The Authors

---

> > ### Comment · Reviewer_6GUt · 2024-11-24
> >
> > Thanks for providing the revision! I have increased the score accordingly.

---

> > > ### Author Response · Authors · 2024-11-24
> > >
> > > Thanks for your encouraging comment. We would like to appreciate again for your valuable suggestions.

---

### Official Review · Reviewer_1ATp · 2024-11-02

**Soundness:** 3
**Presentation:** 3
**Contribution:** 3
**Rating:** 6
**Confidence:** 3

**Summary:**

This paper proposes a novel Mixture-of-Depth (MoD) adaptation method, $\gamma$-MoD, designed to improve computational efficiency for multimodal large language models (MLLMs). By introducing ARank, a metric to identify redundant layers, $\gamma$-MoD selectively replaces dense layers with MoD layers to reduce computational costs while maintaining performance. The model is validated across several benchmarks, showing promising efficiency gains.

**Strengths:**

1. $\gamma$-MoD significantly reduces training and inference time by efficiently identifying and replacing redundant layers.
2. The use of ARank as a layer redundancy metric is novel and provides a structured approach to layer selection in MLLMs.
3. The shared vision-language router and masked routing learning modules are thoughtfully designed for multimodal alignment, enhancing the model's performance in cross-modal tasks.

**Weaknesses:**

1. ARank’s accuracy heavily depends on the diversity and representativeness of the input samples, which may lead to inaccurate redundancy assessment, especially in varied multimodal tasks.
2. ARank’s reliance on low-rank attention maps to identify redundancy may overlook the contextual importance of low-rank layers, which could contain compressed but critical information.

**Questions:**

How does $\gamma$-MoD ensure that ARank’s redundancy metric accurately reflects the importance of layers across diverse multimodal tasks, especially given the unique challenges posed by visual information?

---

> ### Author Response · Authors · 2024-11-22
>
> ------
>
> > **Comment 1: ARank’s accuracy heavily depends on the diversity and representativeness of the input samples, which may lead to inaccurate redundancy assessment, especially in varied multimodal tasks.**
>
>
>
> **Response**:  Thanks for this professional comment.  We fully agree that, as a sample-dependent metric, the estimation of ARank will fluctuate as samples change.  However, when we use a small batch of samples to estimate ARank, the variance of ARank is indeed negligible for the determination of the MoD layer.  Figure 3 in our paper strongly demonstrates this property, where the Arank values of different layers are almost unchanged regardless of samples and tasks.  In this case,  ARank  could  be a stable metric for the choice of MoD layers.
>
>
>
> To further eliminate your confusion, we would like to provide the layer-wise ARank values  of different tasks in the table below, which further suggests the robustness of ARank.
>
> | Task | Layer-8 | Layer-16 | Layer-24 | Layer-32 |
> | ---- | ------- | -------- | -------- | -------- |
> | GQA  | 1044    | 590      | 431      | 930      |
> | SQA  | 1044    | 590      | 431      | 946      |
> | MMMU | 1054    | 542      | 414      | 925      |
>
>
>
> ------
>
> > **Comment 2:  ARank’s reliance on low-rank attention maps to identify redundancy may overlook the contextual importance of low-rank layers, which could contain compressed but critical information.**
>
>
>
> **Response**: Thank you for this insightful advice.   ARank  is defined from the perspective of  layer redundancy, so it does not consider the impact of contextual information.  However, ARank seems to be already a good indicator for preserving important contextual content.   As shown in Figure 4, the visual contents relevant to the question are well preserved while most of the unimportant contents are discarded.
>
>
>
> Nevertheless, we still agree that the consideration of contextual information will be highly beneficial to our methods. In fact, we have already adopted a context-preserving strategy in our method, namely masked routing learning, which aims to preserve significant context (textual instruction) in MoD layer.    Based on your suggestions, we will explore more strategies to combine the context importance with $\gamma-$MoD.
>
>
>
> ---
>
> > **Comment  3:  How does γ-MoD ensure that ARank’s redundancy metric accurately reflects the importance of layers across diverse multimodal tasks, especially given the unique challenges posed by visual information?**
>
>
>
> **Response**: Thanks for this comment.   As discussed in **Comment 1**,  although ARank is a sample-dependent metric, it is quit stable across samples and tasks.  We think that ARank may reflect the inherent redundancy of a given pre-trained model, so that it does not change significantly across samples and tasks. Our visualizations of Fig 3 and the effectiveness of ARank on 9 multimodal benchmarks have reflected this property.  We hope  these discussion can further help you understand the robustness of ARank.
>
> ---

---

> ### Author Response · Authors · 2024-11-24
>
> Dear Reviewer 1ATp,
>
> Thanks again for your great efforts and constructive advice in reviewing this paper!   **After reading our rebuttals, Reviewer 6GUt and wmRq have raised their scores to 6 and 8, respectively. (Reviewer 6GUt also gives the highest score for soundness. )**  To date, all of your and other reviewer's concerns are updated to our new revision. And our source codes and pre-trained weights are  anonymously released at: https://anonymous.4open.science/r/Gamma-MOD-code-AC17.
>
>
> With the discussion period drawing to a close, we are particularly concerned about whether our revisions adequately address your concerns，and sincerely hope to resolve  all potential  issues to earn a more positive rating from you.
>
> Best regards!

---

### Meta-Review · Area_Chair_styv · 2024-12-15

**Metareview:**

This paper proposes γ-MoD, a novel Mixture-of-Depth (MoD) adaptation strategy for multimodal large language models (MLLMs), aimed at improving computational efficiency. By introducing ARank to identify redundant layers, the method effectively reduces training and inference costs while maintaining performance. Strengths include the innovative use of ARank, the practical efficiency gains, and the robustness demonstrated across multiple benchmarks. While some limitations exist, such as the need for more detailed comparisons with alternative sparsity methods and further exploration of routing mechanisms, the revisions adequately addressed reviewers' major concerns. Overall, the work makes a meaningful contribution to efficient MLLM design and is recommended for acceptance as a poster.

**Additional Comments On Reviewer Discussion:**

The reviewer discussion highlighted key concerns, including ARank's robustness, applicability to existing MLLMs, and the need for more comparisons with alternative sparsity methods. The authors provided detailed rebuttals and revisions, which were well-received by the reviewers. Notably, additional experiments on new MLLMs and improved clarity on ARank's stability strengthened the paper's contributions. Reviewers adjusted their ratings positively, reflecting the authors' effective responses.

---

### Decision · Program_Chairs · 2025-01-22

Accept (Poster)